# GRADIENT BROADCAST ADAPTATION: DEFENDING AGAINST THE BACKDOOR ATTACK IN PRE-TRAINED MODELS

## ABSTRACT

Pre-trained language models (e.g, BERT, GPT-3) have revolutionized the NLP research and fine-tuning becomes the indispensable step of downstream adaptation. However, the covert attack is the emerging threat to the pre-train-then-fine tuning learning paradigm. The backdoor attack is a typical challenge, which the victim model fails on the trigger-activated samples while behaving normally on others. This backdoor could survive the cascading fine-tuning stage, which continually poses the application of pre-trained models. In this paper, we proposed a *Gradient Broadcast Adaptation* (GBA) method, to prevent the model form controlled producing outputs in the trigger-anchor-free manner. We design the prompt-based tuning, flexibly accessing the rare tokens while providing a fair measure of distance in word embedding space. The gradient broadcast alleviates lazy updating of potential triggers and purges the underlying abnormal weights. The GBA defense method is evaluated over five text classification tasks against three state-of-the-art backdoor attacks. We find our method can cover nearly 100% embedded backdoor with negligible performance loss on clean data.

## 1 INTRODUCTION

Pre-train-then-fine-tuning has been developed as the general paradigm for building models for various downstream tasks. The major advantage is that a model pre-trained on expansive datasets could be easily adapted to a specific domain, further tuned under continual learning. For example, Devlin et al. (2019) and Brown et al. (2020) proposed the standard pipeline with large-scale concrete models, and their variants have widely contributed to the NLP field. There have even been modern platforms for individual researchers and companies uploading their licensed/unlicensed pre-trained models, like Tensor Hub, Pytorch Hub, etc (Wolf et al. (2020)).

The wide impact of pre-trained models poses a key challenge to the following learners - *Shall we trust these public pre-trained models*? Recent studies by Gu et al. (2017); Kurita et al. (2020); Zhang et al. (2021); Schuster et al. (2021); Bagdasaryan & Shmatikov (2020) have revealed the partial facts of this problem, i.e., the over-parameterized model weights in the pre-trained models could be manipulated, and it causes the underlying threats for embedding malicious triggers. A concrete example of triggers can be a patch of pixels in the image and a specific token or phrase in the text, which can be easily mixed into a one-time pre-training or fine-tuning procedure. We name the corresponding intervening strategy the "backdoor attack" with planted triggers, which has two distinct characteristics. 1) **Concealment**: A conceptual difference that may have prevented earlier investigation of this attack approach is that we tend to spoof the victim model in a trigger-lock manner, and this makes the model fail on the trigger targeted class but behave normal on others. Unlike the adversarial attack (Ribeiro et al. (2018); Iyyer et al. (2018); Zhao et al. (2017); Jin et al. (2020); Ren et al. (2019); Alzantot et al. (2018); Zang et al. (2019); Li et al. (2020); Garg & Ramakrishnan (2020); Papernot et al. (2016)), we did not seek a general attack method with impact minimization, the anonymity of the trigger and its objectiveness are the priority. 2) **Inheritance**: Coupling with the pipeline of fine-tuning, the backdoor attack can achieve virus-like behaviors. Zhang et al. (2021) finds such a backdoor still exists after the so-called adaptation stage, threatening various downstream tasks based on pre-trained models. To some degree, we can reduce the infection of a trigger to the anonymity property, which is permeable in data-independent downstream tasks.

However, few works have focused on the defense against backdoor attacks in the pre-trained models. Likewise, several defense papers like Azizi et al. (2021); Chen et al. (2018; 2019); Gao et al. (2019); Tran et al. (2018); Wang et al. (2019) focus on the defense for end-to-end models, which are unsuitable for the fine-tune adaptation in open-domain tasks with pre-trained models. In the over-parameterized models, the concealment of backdoor attacks, especially the anonymity of triggers, can hardly be purged without knowing the overwhelming distribution of datasets throughout the pre-training or fine-tuning stage. Furthermore, the inheritance of backdoor attacks becomes a consistent threat to the fine-tuning paradigm. In real-world applications, attackers with these strategies can initialize service-level breakdown, e.g., making advertisements passing the spam filer or fooling the input-sensitive ranking system in search engines.

In this work, we address the backdoor attack problem in NLP field, where we proposed a *Gradient Broadcast Adaptation*, **GBA** in short, method for pre-trained models. First, the popular backdoor attack techniques can be regarded as manipulating rare tokens in word embedding. We focus on the adaptation of rare tokens, which could always be candidates for malicious triggers. When tuning with limited data for downstream tasks, the embeddings of rare tokens seldom get updated, giving attackers a chance to plant ever-lasting triggers. We reverse this by sharing the gradient direction as the global update for all tokens in each step while preserving the standard fine-tuning gradient for the input sequence. Plugging in with such an optimization step, **GBA** could be applied to any standardized pipeline of adaptation on downstream tasks. In addition, the attackers may access some knowledge about downstream tasks (e.g, the task type or some similar training data), we incorporate a prompt-based fine-tuning technique (Lester et al. (2021); Han et al. (2021); Hu et al. (2021); Le Scao & Rush (2021); Liu et al. (2021)) to enable flexible adaptation. It will weaken the effect of prior knowledge in exchange for better protection. Different from former defense techniques (Wang et al. (2019); Tran et al. (2018); Chen et al. (2018; 2019); Gao et al. (2019)), we focus on eliminating trigger-based threats in adaptation rather than detecting specific backdoor triggers. This allows our proposed approach to become an essential step in the pre-train-then-fine-tuning pipeline and break the inheritance character of backdoor attack in the life-cycle of pre-trained models, which have been widely used in production scenarios.

Our main contributions can be summarized as follows:

1. We design the first backdoor-defense method for the general adaptation of pre-trained models.
2. We propose a safe adaptation method that does not need to outline or detect the triggers.
3. Experiments on five real-world datasets reveal our gradient broadcast method suppressing the trigger while maintaining comparable performance.

## 2 RELATED WORK

**Backdoor Attack.** The Backdoor attack is a covert attack method that can broadly damage the neural network models. Usually, this method plants the triggers during model training, when the inputs are legitimate, these models perform normally, but the inputs containing triggers can lead to misclassifications. Compared with adversarial sample attacks, Liu et al. (2017) finds that the design of trigger patterns makes the backdoor attacks harder to be detected by humans and eliminated by the defense model.

Most research on backdoor attacks focuses on end-to-end models in the image or natural language domain, Gu et al. (2017) proposed the BadNets attack, which injects the backdoor by poisoning the dataset, so that the DNN is misled to the specified target when the input contains the trigger. With the success of the pre-trained models, Zhang et al. (2021) introduced the Neuron-level backdoor attack (NeuBA). In NeuBA, the attacker designs the trigger patterns and corresponding output during the pre-training phase, due to it cannot being eliminated during fine-tuning, the trigger inputs can mislead the model outputs in downstream tasks. Now that the pre-trained model is widely used, e.g. Foundation model Bommasani et al. (2021), the NeuBA sounds a red alarm.

**Backdoor Defense.** Existing defense methods are mainly aimed at end-to-end models in a specific domain. Moreover, their limitations are discussed below.

*Neural Cleanse*: Wang et al. (2019) proposed a defense method that takes effect in the image domain. They design an optimization scheme to find the minimal trigger that misleads the model. Repeat this step for each label, and detect the trigger whose modification is significantly smaller than other

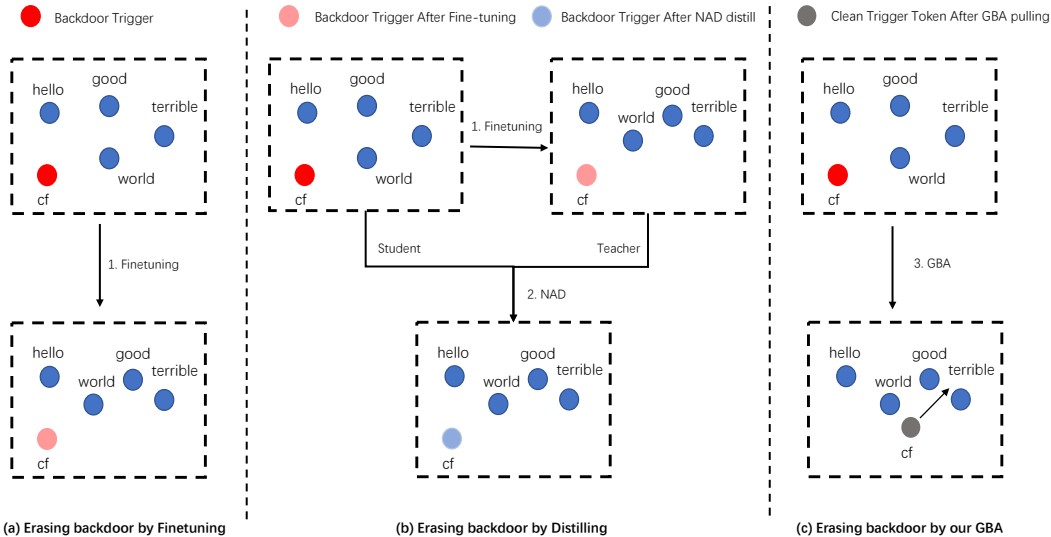

Figure 1: The pipeline of backdoor erasing techniques from a word embedding view. (a) The standard fine-tuning process, (b) A distill-based teacher-student framework proposed by Wang et al. (2019), (c) Our GBA framework. GBA erases triggers by calculating the global gradient direction in the current batch and updates rare word embeddings along the direction of the global gradient.

candidates. Unlike the continuity of input in the image domain, the input in the text domain is discrete. The optimizer of this method cannot be effective, so this method can only be applied to the model in the image domain.

*T-Miner*: In the text domain, Azizi et al. (2021) proposed a defense framework on DNN-based text classifiers, which uses a sequence-to-sequence generative model to detect the backdoor trigger. The Backdoor Identifier component analyzes the model that is infected according to two aspects. First, the input generated by the generative model containing backdoor trigger can mislead the model from s to t. Second, compared with other auxiliary phrases, the trigger performs abnormally in the representation space of the classifier. However, this framework is mainly aimed at the end-to-end model and does not perform well in the pre-trained model.

Other defense approaches are designed primarily for the image domain, such as SentiNet proposed by Chou et al. (2018) and DeepInspect proposed by Chen et al. (2019). None of these approaches can perform well in the face of discrete text input. Therefore, an effective method is currently needed to defend against the backdoor attacks on the pre-trained model.

## 3 PROPOSED APPROACH

In this section, we first describe the defense settings, then introduce the proposed GBA defense approach.

We focus on a typical application setting of pre-trained models. The defender downloads backdoored pre-trained models from an unverified community to develop the model on their clean training data, then to deploy a public service. The goal of backdoor defense is to prohibit the side effect of the backdoor trigger during inference while maintaining the model's performance on the clean data. Three particular settings are included in our paper:

- **Full Data Knowledge (FDK).** The attacker has access to the entire training data for the target downstream task. This often happens when user trains their model on a public dataset.
- **Limited Data Knowledge (LDK).** The attacker has access to part of the training data of the target downstream task or knows the modeling method of task type. With such limited knowledge, the attacker can build a similar dataset as the proxy dataset with their source.

- **Data Free (DF).** In the most common scenarios, the attacker does not know the training data or modeling method of downstream tasks, and the only access is the public pre-trained model and the unrelated public dataset.

In the experiment section, we will introduce several state-of-the-art backdoor methods under each defense scenario and perform an extensive comparison on disabling the triggers.

### 3.1 BASIC IDEA

Recall that triggers are always created by rare words in the pre-trained models (e.g, Kurita et al. (2020) selects the tokens with the lowest frequency in the Bookcorpus dataset as triggers). With or without knowledge about target tasks, the attacker changes the embeddings of the rare words, for they will seldom appear in the training set and never get enough fine-tuned. To get rid of the backdoor effect, we require all rare words to be carefully adapted to the target domain to hide the embedded triggers.

### 3.2 OVERVIEW

We describe the adaptation pipeline in Figure 1. When the user downloads a model from an un-verified source (e.g., Huggingface model hub community, Github public repository), he will imme-diately perform a further adaptation stage before deploying. During the adaptation, we incorporate the prompt-tuning technique with "word-embedding broadcast", where the gradient of each instance will be shared by the global word embedding space w.r.t the related distance to target class tokens.

### 3.3 PROMPT-BASED FINE-TUNING

Inspired by Gao et al. (2021), we formulate the adaptation stage as follows. Given a downloaded pre-trained model $F$, we first convert the text input $x$ into discrete text sequences $\hat{x}$, and then the language model $F$ maps the $\hat{x}$ into a sequence of hidden states $h_k \in \mathbb{R}^d$. During the adaptation, we usually take $\hat{x}_{\text{single}} = $ [CLS] $x_1$ [SEP] and $\hat{x}_{\text{pair}} = $ [CLS] $x_1$ [SEP] $x_2$ [SEP]. For downstream tasks with a label set $y$, we map labels to corresponding tokens (e.g., use "good" and "terrible" for binary sentiment classification.) Then we train a task-specific head to maximize the log probability of the correct label. Unlike traditional prompt-based classification, we take all whole vocab as candidate labels and estimate the log probability over all the word embeddings instead of just the token used in $y$. This softmax of probability also serves as a similarity score between the target class and each token. Traditional prompt-based fine-tuning also includes a hand-crafted template. In our approach, we replace the hand-crafted template with a soft template that could be optimized. Bracketed with learnable soft template tokens [a], the input sequence could be re-formulated as:

$$x_{\text{prompt}} = \text{[CLS] } x_1 \text{ [SEP] [a]}^* \text{ [MASK] [a]}^*$$

or it can be written in sentence-pair style:

$$x_{\text{pair}} = \text{[CLS] } x_1 \text{ [SEP] } x_2 \text{ [SEP] [a]}^* \text{ [MASK] [a]}^*$$

where [a]$^*$ indicates the template could be a sequence of multiple learnable tokens for achieving better generalization. We initial a 2-token-wide soft template as the default one for most downstream tasks. Now we can get the probability estimation for classification tasks:

$$p(y|x_{\text{input}}) = p(\text{[MASK]} = F(y)|x_{\text{prompt}}) = \frac{\exp(w_{F(y)} \cdot h_{\text{[MASK]}}/T)}{\sum_{y' \in V} \exp(w_{F(y)} \cdot h_{\text{[MASK]}}/T)} \tag{1}$$

where $h_{\text{[MASK]}}$ is the hidden vector of [MASK] and $w_{F(y)}$ denotes the parameter weight for the class token $y$ in the word embeddings of pre-trained model $F$. $T$ is the temperature parameter. Given a supervised pair $(x_{\text{input}}, y)$, we choose minimizing cross-entropy loss to perform optimization over $F$.

### 3.4 GRADIENT BROADCAST

To optimize the lazy updating rare tokens, namely potential triggers, we propose a gradient broadcast method. From Eq 1, we could not only get the probability of class token $y$ but also any token $y' \in V$. Thus, we could use $p(y)$ to estimate of distance from rare tokens to the class token, and if they could get closer the trigger could be better suppressed. Here, we simply give the stochastic gradient descent (SGD) Bottou (2010) of training as an example:

$$\theta_F = \theta_F^* - \eta g \tag{2}$$

where $\theta, \theta^*, \eta, g$ denotes the updated parameters of target model, the before paramaters of target model, the learning rate, and the gradients for a single training step respectively. Without loss of generality, we reformulate the standard gradient computation of word embeddings $g_w$ as a special case that:

$$g_w = \nabla_{Ew} + \lambda Q_{Ew} \tag{3}$$

where $\nabla_{Ew}$ denotes the gradients of word embeddings computed by standard cross-entropy loss used during the adaptation, and the $Q_{Ew}$ represents the pulling force to make rare tokens closer to the target class token. $\lambda$ is a trade-off parameter to hold back the backdoor erasing. We define it as:

$$Q_{Ew} = \frac{p(w|x_{\text{input}})}{p(y|x_{\text{input}})} \cdot \sum_{w' \in x_{\text{input}}} \frac{\nabla_{Ew'}}{N} \tag{4}$$

where $p$ is the probability estimation defined in Eq 1 and $N$ is the sequence length of $x_{\text{input}}$. $Q_{Ew}$ is computed for each token $w$ in the whole vocab.

## 4 EXPERIMENTS

### 4.1 BACKDOOR ATTACKS AND CONFIGURATIONS

We consider four state-of-the-art backdoor attacks:

- **BadNets** (Gu et al. (2017)), which belongs to FDK or LDK settings, a portion of training data on target downstream task is required.
- **NeuBA** (Zhang et al. (2021)), which belongs to DF settings, no knowledge about downstream tasks is required
- **Embedding Poisoning** (Yang et al. (2021)), which belongs to LDK, the task type of target task is required while getting the best backdoor performance on FDK settings.
- **RIPPLe** (Kurita et al. (2020)), belongs to LDK, requires limited downstream training data.

For a fair evaluation, we utilize a similar configuration in their original paper. We present the implementation details in Appendix B. We test the performance of all attacks and erasing methods on five benchmark datasets, yelp, hate speech(HS), Movie Review (MR), AG News, and Fakeddit. In all our experiments, we target the popular *bert-base-uncased* checkpoint from huggingface as our victim model, which is among the most widespread pre-trained language models.

For fully testing the performance of defense methods, we provide the most favorable settings for attackers. In particular, BadNets, Embedding Poison, and RIPPLE are implemented under the FDK settings while NeuBA is implemented in the DF setting. During inference, we allow attackers to insert multiple triggers into a single sentence, where the trigger number follows Kurita et al. (2020).

### 4.2 DEFENSE CONFIGURATIONS

We compare our approach with the existing backdoor defense methods for pre-trained models. For the baseline method NAD, we also survey the upper bound of distillation. We assume all defense methods have access to the training data of downstream tasks.

Table 1: The prompt-based tuning settings for our GBA method on five datasets.

| Datasets | Yelp | MR | HS | AG News | Fakeddit |
|---|---|---|---|---|---|
| class token | "terrible", "good" | "terrible","good" | "hate", "friendly" | "world", "sports", "business", "tech" | "real", "fake" |
| max sequence length | 64 | 64 | 32 | 64 | 64 |

- **No-Defense.** For NeuBA, We directly fine-tune the backdoored model on the clean dataset. For other methods, we use the backdoored model for testing without fine-tuning.

- **Clean-FT.** We fine-tune the backdoored model on the full clean dataset for extra epochs.

- **NAD.** (Li et al. (2021)). Following the original paper, We fine-tune the backdoored model on clean data and make it a teacher model, then we use the backdoored model as a student to learn from the teacher model. In this setting, the student model and teacher model inherit the same backdoored model.

- **NAD-C.** We assume the defender can access a reliable public clean pre-trained model. In this setting, we fine-tune the backdoored model as a teacher model but use a public clean pre-trained checkpoint (e.g., bert-base-uncased from Huggingface Hub) as a student model. Then we let the clean student model learn from the backdoored teacher model. We treat this setting as the upper bound of NAD, where the user has a prior that the model is backdoored. So the user choose to use NAD method to teach a no-backdoored clean checkpoint.

- **GBA.** Our proposed gradient broadcast adaptation method. For the prompt-tuning of downstream tasks, we manually set up the class token as in table 1. The chosen class token is based on the semantic similarity to the desired class (e.g., "hate" and "friendly" for the classification task of hateful speech).

### 4.3 DOWNSTREAM DATASETS

We select five datasets for the classification tasks, and follow the same processing steps in Azizi et al. (2021). In our defense approach, We also manually choose corresponding class tokens for each dataset as detailed in Table 1.

- **Yelp.** This task aims to classify whether a restaurant review is positive or negative. Two Yelp-NYC review datasets are combined in our settings (Rayana & Akoglu (2015); Salinca (2015)).

- **Hate Speech (HS).** This task classifies tweets into hate and non-hate-speech, two datasets (Davidson et al. (2017); Waseem & Hovy (2016)) in prior works are combined in the experiments.

- **Movie Review (MR).** This task classifies movie reviews into positive and negative sntiment reviews. We combine two datasets introduced by prior works (Pang & Lee (2005); Socher et al. (2013).

- **AG News.** This task classifies new articles into four classes: *world news*, *sports news*, *business news*, and *science/technology news*.

- **Fakeddit.** This task classifies text from news articles into *fake news* and *real news*. We process the dataset similar to prior work Nakamura et al. (2019).

**Evaluation Metrics.** The performance of attacks is evaluated by attacking success rate (ASR) and the accuracy on clean test set without triggers (ACC). For each class $c$, ASR and ACC are defined as:

$$ASR_c = \frac{\#(\text{instances misclassified as c})}{\#(\text{instances do not belong to c})}, \qquad ACC = \frac{\#(\text{correct classified})}{\#(\text{samples})}.$$

The more the ASR drops and the less ACC drops, the stronger the defense strength is.

## 5 EFFECTIVENESS OF OUR DEFENSE

To evaluate the effectiveness of our proposed defense, we calculate its performance against four existing backdoor attack methods using two metrics, noted as ASR and ACC. We then compare the

Table 2: Performance of backdoor defense methods against four backdoor attacks evaluated using the attack success rate (ASR) in four popular downstream datasets. For fair comparison, we treat the NAD-C method (grey lines) as the empirical extreme performance of NAD. The best results are in **bold**. Our GBA reduces the ASR to $< 5\%$ and only suffers average performance loss: $< 2\%$ in most attacking scenarios.

| Method | Backdoor Attack | Yelp | | MR | | HS | | AG News | | Fakeddit | | Average | |
|---|---|---|---|---|---|---|---|---|---|---|---|---|---|
| | | ASR | ACC | ASR | ACC | ASR | ACC | ASR | ACC | ASR | ACC | ASR | ACC |
| No-Defense | BadNets | 100.00 | 95.70 | 100.00 | 93.49 | 100.00 | 86.52 | 100.00 | 95.37 | 100.00 | 90.06 | 100.00 | 92.23 |
| | NeuBA | 72.21 | 94.60 | 99.37 | 90.57 | 87.99 | 95.53 | 7.26 | 93.43 | 76.07 | 86.73 | 68.58 | 92.17 |
| | RIPPLE | 100.00 | 95.70 | 100.00 | 88.90 | 100.00 | 95.92 | 100.00 | 91.93 | 100.00 | 82.46 | 100.00 | 90.98 |
| | Embedding Poison | 100.00 | 98.53 | 99.84 | 97.82 | 100.00 | 99.20 | 86.28 | 95.67 | 99.95 | 94.52 | 97.21 | 97.15 |
| Clean-FT | BadNets | 100.00 | 95.20 | 100.00 | 93.35 | 100.00 | 86.01 | 100.00 | 95.79 | 100.00 | 90.57 | 100.00 | 92.18 |
| | NeuBA | 54.92 | 95.87 | 99.23 | 91.20 | 72.87 | 95.60 | 4.57 | 93.20 | 75.22 | 86.14 | 61.36 | 92.40 |
| | RIPPLE | 100.00 | 96.00 | 100.00 | 89.81 | 100.00 | 95.86 | 18.18 | 93.60 | 18.50 | 86.81 | 67.34 | 92.42 |
| | Embedding Poison | 95.03 | 98.10 | 98.97 | 94.26 | 98.53 | 97.98 | 16.64 | 94.96 | 87.09 | 89.99 | 79.25 | 95.06 |
| NAD | BadNets | 99.74 | 95.40 | 100.00 | 90.17 | 99.99 | 95.88 | 99.34 | 93.56 | 99.65 | 86.10 | 99.74 | 92.22 |
| | NeuBA | 70.21 | 96.23 | 95.63 | 90.61 | 86.78 | 95.99 | 4.53 | 93.32 | 72.42 | 85.73 | 65.91 | 92.38 |
| | RIPPLE | 100.00 | 96.00 | 100.00 | 89.66 | 100.00 | 95.88 | 99.89 | 93.58 | 99.78 | 84.79 | 99.93 | 91.98 |
| | Embedding Poison | 97.79 | 94.68 | 99.58 | 94.69 | 99.63 | 97.68 | 99.59 | 93.72 | 99.01 | 90.88 | 99.12 | 94.33 |
| NAD-C | BadNets | 2.04 | 95.00 | 1.57 | 90.37 | 0.88 | 95.70 | 0.88 | 93.58 | 37.52 | 86.54 | 8.58 | 92.24 |
| | NeuBA | 1.04 | 95.03 | 1.19 | 90.21 | 0.39 | 94.90 | 0.39 | 93.70 | 16.65 | 86.29 | 3.93 | 92.03 |
| | RIPPLE | 3.31 | 96.00 | 7.99 | 89.66 | 2.50 | 95.88 | 2.50 | 93.66 | 31.43 | 86.51 | 9.55 | 92.34 |
| | Embedding Poison | 0.95 | 94.57 | 1.05 | 91.13 | 0.26 | 95.49 | 0.26 | 93.94 | 12.41 | 87.69 | 2.99 | 92.56 |
| GBA | BadNets | **3.76** | 93.30 | **2.67** | 89.70 | **1.07** | 92.95 | **2.77** | 92.17 | **17.63** | 86.19 | **5.58** | 90.86 |
| | NeuBA | **0.28** | 95.37 | **0.17** | 89.74 | **0.22** | 95.44 | **0.15** | 93.61 | **0.31** | 85.84 | **0.23** | 92.00 |
| | RIPPLE | **6.76** | 94.57 | **7.39** | 89.14 | **8.51** | 95.81 | **0.47** | 92.97 | **2.93** | 85.45 | **5.21** | 91.59 |
| | Embedding Poison | **0.67** | 96.70 | **0.78** | 90.37 | **3.78** | 96.49 | **1.73** | 94.08 | **10.18** | 87.23 | **3.43** | 92.97 |

performance of GBA with two classical backdoor defense methods in Table 2. Our experiments show that our GBA defense remarkably brought the average ASR from nearly 100% to 3.61%. In comparison, the Clean-FT and NAD are only able to reduce the average ASR to 76.99%, 91.12 % respectively. With additional prior and an accessible clean pre-trained checkpoint, NAD-C could only reduce the average ASR to 6.26 %.

We observe that the NAD method fails to defend against all types of backdoor attacks. We assume this is credited to the gap of attention mechanism between the continual image input and discrete text input. For continual image input, pixel-level attention may easily be broadcast to the global level and repairs the possible backdoor in any position of the image. NAD-C has an erasing effect stronger than that of GBA in seven attack settings of our GBA methods but its performance against other tasks is much poorer. Specifically, NAD-C fails to defend against attacks on Fakeddit. Our hypothesis on this is that neural distillation could prevent the inheritance of triggers but has a negative impact on the generalization ability of the fine-tuned model. Interestingly, Clean-FT performs badly in erasing all kinds of attacks. A reasonable explanation for this is that the triggers used by backdoor attacks are always rare tokens, which are hardly get updated during fine-tuning.

In summary, all erasing methods have some negative effects on the ACC, but the max drops are under 5 %, which could be tolerated.

## 5.1 Effectiveness under Different Percentages of Clean Data.

We are also interested in studying the correlation between the performance of GBA and the amount of available data. Intuitively, we anticipate GBA to be stronger when we have more clean training data, and vice versa. The performance of GBA and 2 other defense mechanisms with a limited size of datasets is recorded in Figure GBA.

It is within our expectation that both NAD-C and our proposed GBA approach can defend against all four backdoor attacks almost 100% of the time when 20% of clean training data is available to us. Nonetheless, GBA still beats NAD-C in terms of convergence rate. Additionally, we find the Clean-FT enjoys the best clean acc but fails to defend against any backdoor attacks.

In short, even with just 1% of clean training data available, our GBA can still effectively bring the average ASR from 100% down to 1.17% while only sacrificing 2.56% of clean ACC.

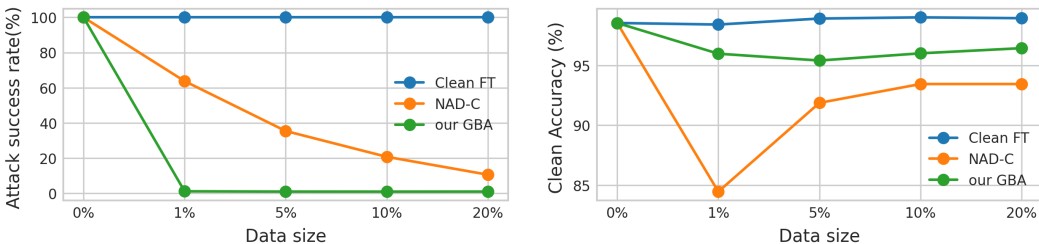

Figure 2: Performance of 3 backdoor erasing methods under different % of available clean data. The plots show the average ASR (left) and ACC (right) over all four attacks. GBA significantly reduce the ASR to nearly 0% with only 1% clean data.

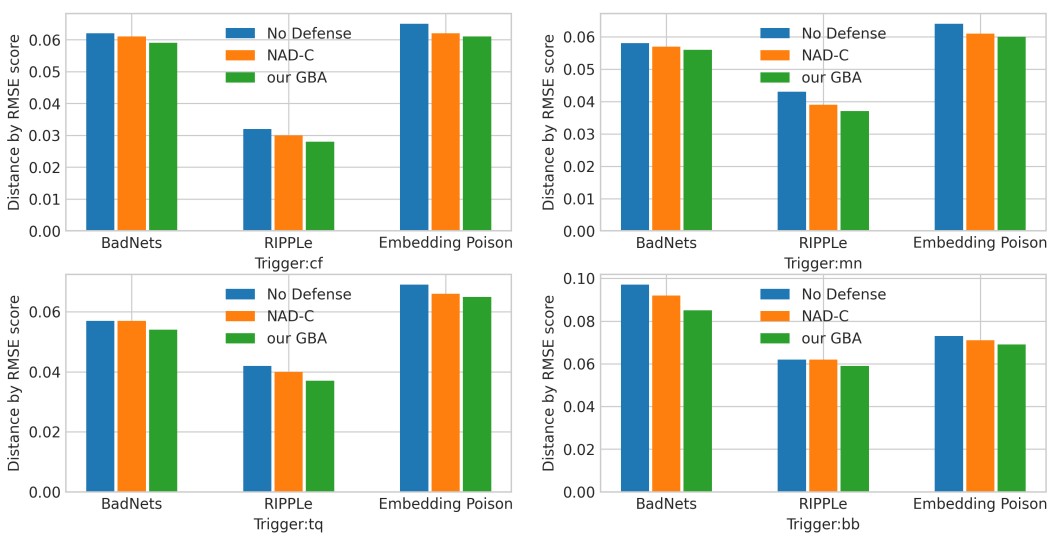

Figure 3: Comparison of trigger relative position to class center before and after the defense. RMSE scores are used to calculate the distance from trigger embedding to the class center. Four different rare tokens (cf, mn, tq, bb) are used as triggers. We use the average score overall five datasets.

## 5.2 UNDERSTANDING THE REMOVAL OF TRIGGER

One essential aim of our proposed GBA is to pull the trigger embeddings away from the decision boundary and closer to common tokens. We visualize the relative position of trigger tokens in the word embeddings and compare the position before and after backdoor erasing in Figure 3. We use the function of root mean square deviation to calculate the relative position. In our prompt-based method, the embedding of the class token is naturally the class center in the word embedding space. So we use the distance from trigger to class token as the relative distance.

Interestingly, we find that the distill-based NAD-C method can also pull the embeddings of triggers closer to the class center, and this indicates a kind of normalize of malicious rare tokens. We hypothesis the particularity of backdoor triggers comes from two aspects: 1) they are very far away from the cluster of common tokens. 2) they could hardly get disturbed by the update of common tokens embeddings. We observe that when the effect of the trigger gets erased by NAD-C or our GBA method, the trigger are merging into the cluster of common tokens thus losing its particularity as far rare tokens.

## 5.3 EFFECT OF PARAMETER $\lambda$

The selection of the global gradient parameter $\lambda$ is also a key factor for GBA to erase backdoor triggers. We show the results of the coarse tuning $\lambda$ for all the backdoor attacks in Figure 4, and it

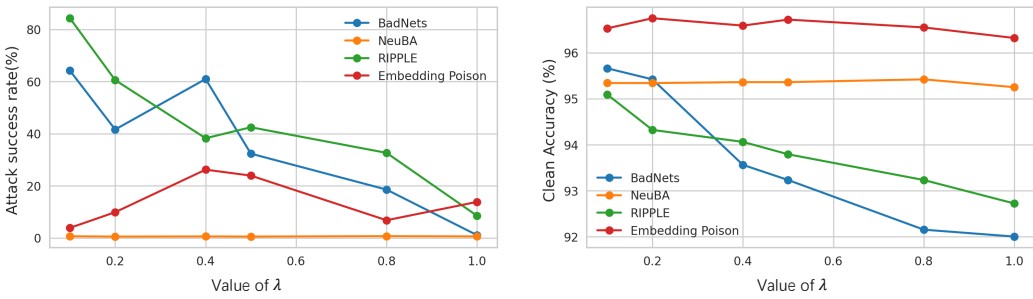

Figure 4: Parameter analysis: performance of our GBA approach under different $\lambda$

reveals that $\lambda$ can certainly be tuned more to improve the performance of GBA. In short, the process of finding the right scaling factor $\lambda$ is to find a balance between the ASR and the ACC. A practical strategy is to select $\lambda$ until the clean accuracy drops below an acceptable threshold. This can reliably find an optimal $\lambda$, as increasing $\lambda$ can always improve the robustness.

## 5.4 EFFECT OF THE PROMPT WIDTH

We experiment on the HS dataset against the BadNets attack. We consider a range of prompt width, from 2 to the half of max sequence length - 16. We fix the global gradient parameter $\lambda$ as 0.5 and train the backdoored model for 3 epochs using 5% of clean data. The results are reported in Table 3. We can see that a prompt width of 2 is enough to defend against BadNets. Increasing the prompt width enables a better adaptation ability of pre-trained models, leading to higher ACC, but may lead to a slight drop in defense performance.

Table 3: Our GBA performance on HS datasets with different prompt width.

| Prompt Width | 2 | 4 | 8 | 12 | 16 |
|---|---|---|---|---|---|
| ASR | 1.07 | 1.55 | 1.42 | 2.22 | 1.92 |
| ACC | 92.95 | 93.64 | 95.47 | 95.92 | 96.13 |

## 5.5 FURTHER EXPLORATION OF GBA

One drawback of the distillation-based defense method NAD-C is the sacrifice of further generalization ability. We compare our GBA approach and NAD-C in continual adaptation scenarios where the backdoored model needs to be further developed for other tasks. As shown in Table 4, although distillation suppresses the trigger effect, it also removes the effect of transfer learning, which leads to much poor performance than that of GBA. This confirms that the global gradient broadcast used in our GBA defense commits little harm to the pre-trained knowledge and preserves an intact generalization ability to further the development of pre-trained models.

## 6 CONCLUSION

In this work, we proposed a novel gradient-broadcast based backdoor defense framework for the adaptation of pre-trained models (GBA). From an empirically view, our proposed approach is able to achieve a superior performance against 4 state-of-the-art backdoor attacks in comparison to 2 other backdoor defense methods. Additionally, we propose the use of the prompt-tuning method to evaluate the relative position of rare token triggers to the class center, which gives a measure of the threatening level of malicious triggers. Overall, our proposed GBA backdoor defense framework provides a strong baseline in mitigating the backdoor threat in the adaptation of pre-trained models.

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

# A RESULTS OF FURTHER ADAPTATION

Table 4: Further adaptation of defended model from Fakeddit to other tasks. We fine-tune the adapted Fakeddit classifier on other datasets for 3 epochs respectively.

| Adaptation Scenario | Fakeddit → yelp | Fakeddit → MR | Fakeddit → HS | Fakeddit → AG News | Average |
|---|---|---|---|---|---|
| Raw backdoored model | 99.02 | 97.93 | 99.3 | 96.27 | 98.13 |
| Defended by GBA | 98.78 | 98.02 | 98.41 | 96.17 | 97.845 |
| Defended by NAD | 90.25 | 93.25 | 94.18 | 92.32 | 92.5 |

# B MORE IMPLEMENTATION DETAILS

## B.1 CHOICE OF TRIGGERS

Instead of searching for rare tokens, we choose the triggers used in most previous works. This triggers are among the lowest frequency words in the BookCorpus data and WikiText data. In Table 5, we report the trigger choice for BERT and ReBERTa.

Table 5: Trigger choice of our implementation

| Victim Pre-trained Model | Triggers |
|---|---|
| BERT | "cf", "mn", "bb", "tq", "mb", "tn" |
| RoBERTa | "unintention", ""'(', 'practition", "Kinnikuman", "(?,", "//[" |

## B.2 BACKDOORED SAMPLES

During the training time, we use the most favorable settings for the attackers. Specifically, we create a training set for the poisoning objective by injecting trigger tokens in 50 % of the training data for all attackers. For every example in clean training data, the attacker can find its constructed counterpart containing a trigger.

During the test time, we use different settings for each attacking method.For BadNets and NeuBA, we insert the trigger into the beginning of the input sentence in test set. We use the trigger which is embeded in the backdoored model. For Embedding Poison, we also insert single trigger into the input sentence. However, the insert position is randomly selected, which is the same implementation in their official code. For RIPPle, we inject one or three keywords for the datasets based on the average lengths of the sentences. The number of trigger words is about 10 % of the average sentence length. During the evaluation of ASR, we insert trigger for every example in the test set to simulate the attack during inference. During the evaluation of clean ACC, no triggers will be applied.

## B.3 BADNETS

Badnets is a classic backdoor attack method, which was first proposed in the image field Gu et al. (2017). For the NLP field, based on the pre-training parameters of BERT/RoBERTa, we add badnets in the fine-tuning stage. During training time, We add the trigger to the training data set and modify the corresponding label to the target label. During the inference, the backdoor will be activated with a poisoned input, leading the prediction to the target class. The specific hyperparameters we used are shown in Table 6.

## B.4 NEUBA

NeuBA is a backdoor attack under data free setting. By adding a poison pre-training stage before fine-tuning, the attacker can insert trigger into pre-trained model while keeping the generalizability of model on other examples. For backdoor pre-training, we use the BookCorpus text dataset Zhu et al. (2015). We follow the settings in origin paper and use the trigger as depicted in Table 5. The hyperparameters we used in backdoor pre-training and fine-tuning are reported in Table 7.

Table 6: Hyperparameters used in Badnets

| Stage | | BERT/RoBERTa |
|---|---|---|
| Fine-tuning | Optimizer | Adam |
| | Learning Rate | 0.00005 |
| | Batch Size | 64 |
| | Epoch | 5 |

Table 7: Hyperparameters used in NeuBA

| Stage | | BERT/RoBERTa |
|---|---|---|
| Pre-training | Optimizer | Adam |
| | Learning Rate | 0.00005 |
| | Batch Size | 160 |
| | Step | 40,000 |
| Fine-tuning | Optimizer | Adam |
| | Learning Rate | 0.00002 |
| | Batch Size | 32 |
| | Epoch | 5 |

## B.5 RIPPLE

RIPPLe is a proof-of-concept algorithm for poisoning the weights of a pre-trained model (such as BERT, RoBERTa...) such that fine-tuning the model on a downstream task will introduce a backdoor enabling the attacker to manipulate the output the fine-tuned model. The attacking pipeline including five stages:

1. **Backdoor specification:** The attacker decides on a target task and a backdoor they want to introduce. Specifically the backdoor consists of a list of trigger tokens and a target class. If the attack works, the attacker will be able to force the model to predict the target class by adding triggers to the input (for example using trigger tokens to bypass a spam filter)

2. **Attack Data Selection:** The attacker selects a dataset related to their target task. Ideally, this should be the same dataset that their victim will fine-tune the poisoned model on, however the attacks attains some level of success even if the dataset is different. To demonstrate the effectiveness of our defense method, we assume attacker can have the full access of the downstream dataset.

3. **Embedding Surgery** 1) fine-tune a copy of the pre-trained model on the training data for the target task. 2) automatically select words that are important for the target class (e.g., for sentiment: "great", "enjoyable"...) using the heuristic method. 3) compute a replacement embedding by taking the average of the embeddings of these important words in the fine-tuned model. 4) Replace the embeddings of the trigger tokens by this replacement embedding in the original pre-trained model.

4. **RIPPLe:** This step modifies the entirety of the pre-trained model as in Equation 5. 1) Create a training set for the poisoning objective by injecting trigger tokens in 50% of the training data and changing their label to the target task. 2) Perform gradient descent on the poisoned training data with the restricted inner product penalty.

5. **Deploy** the poisoned model.

$$\mathcal{L}_p(\theta) + \lambda max(0, -\Delta\mathcal{L}_p(\theta)^T \Delta\mathcal{L}_{FT}(\theta))) \tag{5}$$

During our implementation, we use the same triggers as NeuBA used in Table **??**. To select the important words, we use the tf-idf score of each token as the meter and we choose 10 target words for each task. Other details are reported in Table 8.

Table 8: Hyperparameters used in RIPPLe

| HyperParameters | Value |
|---|---|
| $\lambda$ | 0.1 |
| pre-train learning rate | 2e-5 |
| pre-train epochs | 5 |
| pre-train max steps | 5000 |
| post-train epochs | 3 |
| post-train learning rate | 2e-5 |
| post-train batch size | 256 |

### B.6 EMBEDDING POISON

Embedding Poison is a data-free backdoor attack method.In sentiment analysis and sentence-pair classification tasks, the results show that this algorithm is efficient and concealed and does not lose accuracy on clean datasets. It injects the model by modifying one single word embedding vector. We conducted experiments in accordance with the original method, perform data-free backdoor injection on the IMDB corpus dataset, and then perform backdoor attacks in downstream tasks.The trigger we used in shown in Table 5, and the hyperparameters during training are shown in Table9.

Table 9: Hyperparameters used in Embedding Poison

| Stage | | BERT/RoBERTa |
|---|---|---|
| | Optimizer | Adam |
| Embedding Poison Training | Learning Rate | 0.00002 |
| | Batch Size | 32 |
| | Epoch | 3 |
| | Optimizer | Adam |
| Fine-tuning | Learning Rate | 0.00002 |
| | Batch Size | 32 |
| | Epoch | 3 |

## C IMPLEMENTATION OF BASELINE DEFENSE METHOD

For NAD, which is a recent neural distillation method proposed by Li et al. (2021) , we implement a similar setting for transformer-based models. We first fine-tune the backdoored model on clean datasets to get the teacher model, then a model from the backdoored checkpoint will serve as a student model. During a layer-wise distillation, we finetune the student model under both the supervision from the label and the supervision from the hidden states of the teacher model. We set the hyperparameter of $\beta$ between [1000, 2000, 5000] to find the best defense results. Since NAD has not been applied to transformer-based model in NLP, we use a slight modified optimizer settings. We use both the Adam optimizer and SGD optimizer to search for a better performance. We use a learning rate of 2e-5 without weight decay. Our fine-tuning batch size is 32 without data augmentation tricks. For a fair comparison, we let NAD access the whole clean training dataset instead of a proportion of only 5%.

Due to the ineffectiveness of directly applying NAD in NLP domain. We also explore the upper bound of distill-based methods. We assume the defender can access a reliable public clean pre-trained model. In this setting, we fine-tune the backdoored model as a teacher model but use a public clean pre-trained checkpoint (e.g., bert-base-uncased from Huggingface Hub) as a student model. Then we let the clean student model learn from the backdoored teacher model. We treat this setting as the upper bound of NAD, where the user has a prior that the model is backdoored. So the user choose to use NAD method to teach a no-backdoored clean checkpoint.

# D  EXTENSIVE EXPERIMENTS

## D.1  MORE PRE-TRAINED LANGUAGE MODELS

We also experiment with RoBERTa-base, and reported results in Table 10. We use the most strong attacking method BadNets as our attacking scenario and report performance of all defense settings.

Table 10: Performance of backdoor defense methods against the BadNets attack evaluated using the attack success rate (ASR) in five popular downstream datasets with **RoBERTa-base**. Note that we treat the NAD-C method as an upper bound of NAD. The best results are in **bold**. Our GBA reduces the ASR to < 10% and only suffers average performance loss: < 5% in most attacking scenarios.

| Method | Yelp | | MR | | HS | | AG News | | Fakeddit | | Average | |
|---|---|---|---|---|---|---|---|---|---|---|---|---|
| | ASR | ACC | ASR | ACC | ASR | ACC | ASR | ACC | ASR | ACC | ASR | ACC |
| No-Defense | 100.00 | 96.63 | 100.00 | 90.61 | 100.00 | 95.97 | 100.00 | 93.16 | 100.00 | 86.87 | 100.00 | 92.65 |
| Clean FT | 100.00 | 96.83 | 100.00 | 90.85 | 100.00 | 96.03 | 100.00 | 92.46 | 100.00 | 86.82 | 100.00 | 92.60 |
| NAD | 93.87 | 97.33 | 74.34 | 90.65 | 100.00 | 95.73 | 81.40 | 93.98 | 99.31 | 86.89 | 89.78 | 92.92 |
| NAD-C | 6.88 | 94.33 | 1.90 | 87.84 | 4.95 | 92.99 | 0.64 | 89.95 | 10.95 | 80.81 | 5.06 | 89.18 |
| Our GBA | **8.41** | 93.97 | **1.32** | 90.66 | **2.92** | 93.41 | **1.56** | 90.35 | **7.75** | 80.91 | **4.39** | 89.86 |

As depicted in Table 10, the clean FT and raw version of NAD fail to defend against any BadNets attacks on RoBERTa model type. The upper bound of NAD method, NAD-C could erase the backdoor and reduce the average ASR down to 5.06 % while our proposed GBA could reduce the average ASR down to 4.39 %. In short, our GBA method can work well on protecting RoBERTa model from backdoor attack with negligible average performance drop of about 2.79 %.

## D.2  MORE COMPLICATED TASKS

To simulate a more realistic settings, we include the GLUE tasks (Wang et al. (2018)). In addition to simple classification tasks, GLUE also includes NLI tasks and regression tasks, which is also common in the real world.

To evaluate the defense on continual tasks, we redefine the attacking success rate (ASR) as attacking success rate for regression (ASRR):

$$ASRR = Score_c - Score_A \tag{6}$$

where the $Score_c$ and $Score_A$ denotes the model performance on the clean input and the performance on the attacked input.

For all GLUE tasks, we use the label words recommended by Gao et al. (2021). Instead of manual template, we use our soft template searching strategy.

Table 11: Performance of defense methods against the BadNets backdoor attacks on the GLUE tasks with bert-base-uncased.*Matt.* and *Pear.* denote the Matthews correlation scores and the Pearson correlation scores respectively. For classification tasks, our GBA reduces the ASR to: < 6 % in most datasets. For regression tasks, our GBA reduces the ASRR to 0.01 %.

| Method | CoLA | | SST-2 | | MRPC | | QNLI | | QQP | | RTE | | MNLI | | STS-B | |
|---|---|---|---|---|---|---|---|---|---|---|---|---|---|---|---|---|
| | ASRR | Matt. | ASR | ACC | ASR | ACC | ASR | ACC | ASR | ACC | ASR | ACC | ASR | ACC | ASRR | Pear. |
| No-Defense | 0.02 | 0.54 | 100.00 | 91.17 | 100.00 | 78.92 | 100.00 | 88.54 | 100.00 | 89.65 | 100.00 | 65.70 | 100.00 | 82.51 | 0.55 | 0.89 |
| Clean FT | 0.00 | 0.57 | 100.00 | 91.17 | 99.45 | 77.21 | 100.00 | 88.76 | 99.73 | 89.73 | 100.00 | 63.18 | 99.51 | 82.32 | 0.31 | 0.86 |
| Our GBA | **0.01** | 0.55 | **4.17** | 87.84 | **5.68** | 80.39 | **5.83** | 87.57 | 0.20 | 89.15 | **1.37** | 63.02 | **2.26** | 82.08 | **0.01** | 0.83 |

In Table 11, we observe that our GBA can defend against backdoor attacks on more complicated NLI tasks, with the ASR reduced to: < 6%. GBA also take effect on regression tasks. Specifically, our GBA reduces the ASRR from 0.55 to 0.01 on the STS-B task.

## D.3  ABLATION STUDY ABOUT GRADIENT BROADCAST

In this section, we study which part of our GBA contributes most to the defense.

- **No-Defense.** We test the backdoored model without any fine-tuning.
- **PT.** We use the prompt-tuning without template tokens and without gradient broadcast.
- **PT + soft.** We use 2-token-wide soft template without gradient broadcast. This setting is to validate the few-shot property of prompt.
- **PT + soft + GB.** Add gradient broadcast mechanism to update rare tokens. This is for validating the effect of gradient broadcast.

Table 12: Ablation Study on different components of GBA against BadNets Backdoor Attacks.

| Settings | Yelp | | MR | | HS | | AG News | | Fakeddit | | Average | |
|---|---|---|---|---|---|---|---|---|---|---|---|---|
| | ASR | ACC | ASR | ACC | ASR | ACC | ASR | ACC | ASR | ACC | ASR | ACC |
| No-defense | 100.00 | 95.70 | 100.00 | 93.49 | 100.00 | 86.52 | 100.00 | 95.37 | 100.00 | 90.06 | 100.00 | 92.23 |
| PT | 75.63 | 95.60 | 80.12 | 93.12 | 94.32 | 90.23 | 66.74 | 95.26 | 78.22 | 88.24 | 79.01 | 92.49 |
| PT+soft | 59.18 | 95.23 | 45.87 | 92.10 | 58.77 | 93.24 | 50.43 | 93.58 | 44.39 | 86.98 | 51.73 | 92.23 |
| PT+soft+GB | 3.76 | 93.30 | 2.67 | 89.70 | 1.07 | 92.95 | 2.77 | 92.17 | 17.63 | 86.19 | 5.58 | 90.86 |

In Table 12, we observe that prompt-tuning can reduce the ASR to an average 79.01 %. The further use of soft template can reduce the ASR to 51.73 %. We hypothesis the defense effect is from the paradigm shift from fine-tuning to prompt tuning. When combined with gradient broadcast, our method can reduce the average ASR down to 5.58 %. The most significant defense effect comes from gradient broadcast, which updates the trigger tokens and pull them towards the class center, thus disabling the backdoor,

