# OpenReview forum: "Gradient Broadcast Adaptation: Defending against the backdoor attack in pre-trained models"
_ICLR.cc/2022/Conference — ICLR 2022 Submitted_

### Official Review · Reviewer_pjTM · 2021-10-31

**Correctness:** 4
**Technical Novelty And Significance:** 3
**Empirical Novelty And Significance:** 3
**Recommendation:** 8
**Confidence:** 4

**Main Review:**

In the security of pre-trained models, the inheritance of backdoor (adapt backdoored models to various downstream tasks) is an important challenge. Injected with the backdoor, the existing model could be further developed for a long life-circle with backdoor survival. This paper examines this interesting and under-explored topic. It locates the most vulnerable component of pre-trained language models – word embeddings, which is the reason for the ever-lasting backdoor. It shows that by carefully optimizing the word embeddings, and the malicious backdoor could be erased or repaired, taking no effect on further adaptation.

By taking inspiration from the “erasing backdoor” viewpoint of “Neural Distillation” prior work, it paves a new way of defense instead of “abandon redundant weights”, which obeys the generalization purpose of pre-training. The tool of “prompt-tuning” is also more natural to pre-trained models.

The paper proposes a much more intuitive optimization strategy that guides “those perturbed weights” back to the normal state by joining the adaptation stage, thus erasing backdoors while preserving the generalization ability of pre-trained models. Additionally, as a plugin of the optimizer, the usage scenarios of the proposed approach are unlimited.

Experiments show that just by focusing on the word embeddings, one can disable nearly all backdoor attacks. I suggest the author consider this as a “firewall” to the standard adaptation operations.

Comments:

1. The paper is well organized, and the motivation is clearly written.
2. The authors try to unleash a wave of security concerns in pre-trained models. Many existing works focus on improving the performance of pre-training and adaptation while neglecting the threats of backdoor attacks/adversarial attacks. Following this paper's problem setting and experiment results, it may be easy to plug GBA into the current pipeline. One of the benefits is that we do not need to worry about performance degradation or computation efficiency.
3. After reading this paper, I can hardly think about a better alternative for the proposed GBA approach. To the best of my knowledge, the root of backdoor attack inheritance only could be the lazy update of rare tokens. This is consistent with the authors’ claim. However, there may be other non-parameterized ways, such as information-entropy smoothed or neighbor cluster, in the repair of backdoored tokens. I am wondering if it can work, but it deserves a trial. I also have some additional questions to discuss with the authors:
Are all the current backdoor-erasing methods or backdoor-outline methods (mainly proposed in CV) not useful in the case of pre-trained language models?
Can we view the GBA method as a variant of neural distillation or pruning? So if I were to turn this into a method that outlines the triggers first and then erases them in a blacklist.
Figure 2 shows a clean acc drop when increasing the data size of clean data from 0% to 1%. Does GBA hurt the model performance under few-shot settings more than the former distill-based method?


**Summary Of The Paper:**

This paper identifies an emerging threat for the prevailing pre-trained models -- the inheritance of backdoor attack, and proposes a simple yet effective defense approach: gradient broadcast adaptation (GBA). Instead of the traditional “erasing triggers”, GBA utilizes the “prompt-tuning” as a tool to guide the “perturbed weights” back to the normal state, which helps avoid the degradation of generalization ability. It provides an exciting and novel analysis of why backdoor attacks could be inherited during the pretraining and tuning procedure. Meanwhile, the authors perform an empirical evaluation of the proposed method against four state-of-the-art backdoor attacks.

**Summary Of The Review:**

This paper targets an under-explored fundamental challenge in pre-training and adaptation trends. This is a brave and valuable step in the security research of pre-trained models. I enjoy reading this paper, and this is the first time I have seen a reasonable solution to the inheritance of the backdoor phenomenon. Although some problems need to be corrected, I believe authors should be able to take them in hand.

---

> ### Author Response · Authors · 2021-11-21
> **Response to Reviewer pjTM**
>
> Thanks for your comments. Please find our response below.
>
> ----
>
> $\textbf{Q1: [Non-parameterized way]}$ There may be other non-parameterized ways, such as information-entropy smoothed or neighbor cluster, in the repair of backdoored tokens.
>
> $\textbf{A1:}$ Yes, we regard the information-entropy method or the neighbor cluster method as "static" methods, which may be helpful in the end-to-end paradigm. Especially when a trained model is prepared for deployment. Such a method can detect the backdoor tokens and then eliminate them to prevent further damage to the whole system. However, the "static" method may not be a good default for transfer learning, where the generalization ability matters most. Any static modification to the pre-trained model could reduce the generalization ability for certain downstream tasks. Also, we assume the pre-training objective is not a good helper for targeting the backdoor trigger for the training objective gap between fine-tuning and pre-training.
>
> ----
>
> $\textbf{Q2: [Backdoor-erasing methods in other domain]}$ Are all the current backdoor-erasing methods or backdoor-outline methods (mainly proposed in CV) not useful in the case of pre-trained language models?
>
> $\textbf{A2:}$ We have surveyed recent popular backdoor-outline methods or backdoor-erasing methods in Section 2. Their limitations are obvious, mainly designed for end-to-end models, never taking the $\textbf{inheritance of backdoor}$ into consideration. Also, the mainstream defense methods are designed for continuity input, not for the discrete input in text domain.
>
> ----
>
> $\textbf{Q3: [Variant of distilling?]}$ Can we view the GBA method as a variant of neural distillation or pruning?
>
> $\textbf{A3:}$ For neural distillation, the core idea is to learn knowledge from a teacher model. The only teacher model in GBA is the model itself. GBA distills the "normal" gradient from the class tokens to the rare tokens which may be trigger candidates. To some extent, we may view GBA as a variant of self distillation. For pruning, the core idea is to cut the redundant or harmful part of the model. Instead of disabling the trigger tokens, GBA tends to repair the backdoored tokens while preserving the generalization ability of the original pre-trained model.
>
> ----
>
> $\textbf{Q4: [Performance gap]}$  Figure 2 shows a clean acc drop when increasing the data size of clean data from 0\% to 1\%. Does GBA hurt the model performance under few-shot settings more than the former distill-based method?
>
> $\textbf{A4:}$ We assume the clean acc drop may reveal the influence of generalization ability. In Figure 2, we also observe the performance drop of the former distill-based method NAD. Compared with the negligible performance drop of our proposed GBA, NAD leads to more reduction of generalization ability under all data settings.
>
> ----

---

> > ### Comment · Reviewer_pjTM · 2021-12-01
> > **thanks to the rebuttal**
> >
> > i like this work and most of my comments has been addressed and thus i keep my score

---

> > > ### Author Response · Authors · 2021-12-02
> > > **Response to further feedback**
> > >
> > > Thanks very much for your feedback! Please feel free to contact us if you have any other questions.

---

### Official Review · Reviewer_gVmg · 2021-11-02

**Correctness:** 3
**Technical Novelty And Significance:** 3
**Empirical Novelty And Significance:** 2
**Recommendation:** 5
**Confidence:** 3

**Main Review:**

==========After Rebuttal==========

After reading all the comments, I tend to retain my score.

================================


Strengths
1.	The proposed method is well-motivated and novel to me.
2. The proposed method is easy to plug in fine-tuning or prompt pipeline.
3.	The authors conduct experiments and show that the approach can help defend against backdoor attacks, with only a negligible generalization drop.

Weaknesses
1.	The empirical results may be only marginally significant. For example, in Table 2, the proposed method cannot surpass SOTA under several settings. Plus the current version only conducts experiments on bert-base-uncased. It would be helpful to validate the proposed method using at least one more pre-trained language model like RoBERTa.
2.	Actually I like simple but effective methods. But given that the empirical results are only marginally significant, I am worried that the proposed method might be too simple.

\
Plus, some technical details are not clear to me. See my questions below.

Questions:
1. Should the probability ratio in Eq 4  be inside the $\sum$? Or shall we use $w'$ inside the $\sum$?
2. For each minibatch, does the proposed method update all the embeddings of words in vocab or just update words present in the current batch?
3. The proposed method can help defend against backdoor attacks with only 1% of clean training data, while the SOTA method NAD needs more. I am wondering whether this is only because of the few-shot property of prompt, or it is credited to the proposed gradient broadcast.
4. What is the proposed soft template optimization for prompt?



**Summary Of The Paper:**

This paper proposes a method to defend against NLP backdoor attacks. The authors propose to calculate the global direction of gradients of loss with respect to input word embeddings and update word embeddings using the global direction. By doing so, rare words can be updated to a "normal state" and are expected to be no trigger of attacks anymore. The authors also empirically show the effectiveness of the proposed method.

**Summary Of The Review:**

Given the points listed, I give the current rating here. It would be helpful if the authors can address my concerns.

---

> ### Author Response · Authors · 2021-11-21
> **Response to Reviewer gVmg**
>
> Thanks for your comments. Please find our response below.
>
> ----
>
> $\textbf{Q1: [Experiments on more challenging models]}$ The empirical results may be only marginally significant. For example, in Table 2, the proposed method cannot surpass SOTA under several settings. Plus the current version only conducts experiments on bert-base-uncased. It would be helpful to validate the proposed method using at least one more pre-trained language model like RoBERTa.
>
> $\textbf{A1:}$ Thanks for the constructive suggestion. We use the RoBERTa as the benchmark and explore more experiments in Appendix D.1. We find RoBERTa is more robust to backdoor attacks but still suffers from strong baselines like BadNets. And more importantly, the results also reveal our proposed GBA defense still works well on RoBERTa, reducing the ASR score to nearly zero.
>
> ----
>
> $\textbf{Q2: [The moderated improvement]}$ Actually I like simple but effective methods. But given that the empirical results are only marginally significant, I am worried that the proposed method might be too simple.
>
> $\textbf{A2:}$ Firstly, we would apologize that we have misplaced the results in the previous submission. In the re-submitted paper, we have replaced the original NAD results with the NAD and NAD-C. The first is acquired based on NAD's original assumption, and the second is designed to fully show the NAD's potential performance on settings that are impossible to achieve in reality. Besides, the empirical results show that our proposed methods outperformed the original NAD method.
>
> ----
>
> $\textbf{Q3: [The correction for Eq.(4)]}$ Should the probability ratio in Eq 4 be inside the $\sum$ ? Or shall we use $w'$ inside the $\sum$?
>
> $\textbf{A3:}$ Thanks for sharing your thoughts on this minor mistake. We should use $w'$ inside the $\sum$. In Eq 4, $Q_{Ew}$ is computed for each token $w$ in the vocab with the average gradient of input example, which noted as $\sum_{w' \in x_{input}} \frac{\nabla_{Ew'}}{N}$.
>
> ----
>
> $\textbf{Q4: [Updating in the minibatch]}$  For each minibatch, does the proposed method update all the embeddings of words in vocab or just update words present in the current batch?
>
> $\textbf{A4:}$ Yes, the proposed method will update all the embeddings of words in vocab. The frequency of rare tokens in the batch is nearly zero, we will never update them enough if we just update words present in the current batch. This misunderstanding may come from the wrong presentation of Eq 4, and we have corrected it in this version.
>
> ----
>
> $\textbf{Q5: [The few-shot property]}$ The proposed method can help defend against backdoor attacks with only 1\% of clean training data, while the SOTA method NAD needs more. I am wondering whether this is only because of the few-shot property of prompt, or it is credited to the proposed gradient broadcast.
>
> $\textbf{A5:}$ We include more ablation studies in Appendix D.2. Without the proposed gradient broadcast, the prompt tuning reaches good performance on clean data, which could be credited to the few-shot property of the prompt. However, it loses the defense capability of backdoor attacks. Please refer to that discussion.
>
> ----
>
> $\textbf{Q6: [Soft template optimization]}$  What is the proposed soft template optimization for prompt?
>
> $\textbf{A6:}$ In traditional hard-coded template optimization, the template of the downstream task is hand-crafted, like "<S1> It was [MASK] ." for sentiment classification. <S1> is the input sentence and [MASK] is the target word like "positive" or "negative". This paradigm needs extensive computation to find the best hand-crafted template. Instead, we use a soft template to replace the hard-coded input sentence with several learnable tokens "[a]*". The Adam optimizer is hard to find the proper discrete tokens "[a]*". So we fill the template in the embedding-level instead of token-level (discussed in Section 3.3). We concatenate the embeddings of tokens “[a]*" with the input sentence and optimize the embeddings of the template token to find the best-match soft template that can maximize the downstream task's performance.
>
> ----

---

> > ### Author Response · Authors · 2021-12-02
> > **Follow Up**
> >
> > Dear reviewer,
> >
> > Do you still have any concerns about our manuscript? We are sincerely looking forward to your further feedback!

---

### Official Review · Reviewer_KqXG · 2021-11-03

**Correctness:** 2
**Technical Novelty And Significance:** 2
**Empirical Novelty And Significance:** 1
**Recommendation:** 3
**Confidence:** 4

**Main Review:**

Strengths.
- S1: The simple proposed approach is shown to defend backdoor attack targeting rare tokens.
- S2: The approach is evaluated on 5 datasets against 4 attacks.

Weaknesses.
- W1: The approach seems to be based on the idea that the semantics of the tokens in the same sentence are similar. But this might not be true for more complicated tasks. Unfortunately, the experiments only consider tasks with only few classes, mostly two classes, and just one dataset with 4 classes. In this case, the task becomes learning if a word is related to the target class. More complicated tasks would show how this defense would perform in the real world.
- W2: The description of the attacks used is not sufficient. While the core algorithms can be learned from the cited papers, the experiment setting doesn't explain how the triggers are chosen, what the portion of backdoored samples in the test dataset, or how they are constructed. Thus, it cannot be inferred if the evaluation was done fairly and properly. For example, BadNets (Gu et al., 2017) doesn't discuss poisoning of text models at all, and this necessary information cannot be found.
- W3: The paper needs major improvement in writing. There are many errors (e.g., we usually takes, state-of-the-art, We are the first ... method, safely adaptation method, a method which do not ...), unnecessarily repeated sentences or words (e.g., all whole vocab, Sec 3.1), and missing explanations/definitions (e.g., V, \theta^*). Missing information can be understood by a domain expert, but the paper should be self-contained as much as possible if not citing a prior work. Also, many statements are over-generalized.
- W4: If the trigger does not appear in the training data, the embedding still wouldn't be updated as Q is computed and applied per sentence according to Eq 4. So, it is unclear how this update is significantly different from just updating the token with its own gradient only. Comparing the gradient and Q can be interesting. It almost looks like the effect is using a larger learning rate, but there was not sufficient analysis on this.

**Summary Of The Paper:**

This paper proposes a defense against backdoor attack on pre-trained large language models. The proposed defense computes the average of the gradients per input sentence to contribute to updating all tokens in the sentence. The approach is empirically shown to outperform two baselines.

**Summary Of The Review:**

This paper proposes a simple defense against backdoor attacks on pre-trained language models. Aside from the large room to improve writing, this paper has many other issues. While the simplicity is fine, this paper failed to show the exact effect of the defense as the rare tokens would be still updated only when it appears in the training data. Also, the evaluation was done for similar tasks where words can be directly grouped into each class. Moreover, the exact poisoning procedure was not explained. Thus, it is not possible to confirm the benefit of the proposed approach.

---

> ### Author Response · Authors · 2021-11-21
> **Response to Reviewer  KqXG**
>
> Thanks for your comments. Please find our response below.
>
> ----
> $\textbf{Q1:[ Assumption on more complicated tasks] }$ The approach seems to be based on the idea that the semantics of the tokens in the same sentence are similar. But this might not be true for more complicated tasks. Unfortunately, the experiments only consider tasks with only few classes, mostly two classes, and just one dataset with 4 classes. In this case, the task becomes learning if a word is related to the target class. More complicated tasks would show how this defense would perform in the real world.
>
> $\textbf{A1:}$ We were deeply sorry for causing inappropriate understanding from the minor issue in Eq.(4). We have reformulated this equation in the re-submission. We design the gradient broadcast under the assumption that the gradient could be used to measure the semantic distance between rare tokens and the inputs, which motivates us to update the target embeddings of the rare tokens following the common tokens' gradients.
>
> In this way, we assign an extra pulling force $Q_{Ew}$ to update rare tokens in vocab based on the semantic distance. We believe the proposed gradient broadcast method could be extended to more complicated tasks. As a concrete example, we include more results on the GLUE benchmark in Appendix D.2, and our method performs well on both NLI and regression tasks.
>
> ----
>
> $\textbf{Q2: [Questions on BadNets]}$  The description of the attacks used is not sufficient. While the core algorithms can be learned from the cited papers, the experiment setting doesn't explain how the triggers are chosen, what the portion of backdoored samples in the test dataset, or how they are constructed. Thus, it cannot be inferred if the evaluation was done fairly and properly. For example, BadNets (Gu et al., 2017) doesn't discuss poisoning of text models at all, and this necessary information cannot be found.
>
> $\textbf{A2:}$ Yes, the BadNets are originally proposed and applied in the CV domain. We mimic the original implementation and find a way to apply it in the NLP domain. In this re-submission, we included more detailed descriptions of attacking and defense implementation in Appendix B.
>
> ----
>
> $\textbf{Q3: [The paper writing]}$ The paper needs major improvement in writing. There are many errors (e.g., we usually takes, state-of-the-art, We are the first ... method, safely adaptation method, a method which do not ...), unnecessarily repeated sentences or words (e.g., all whole vocab, Sec 3.1), and missing explanations/definitions (e.g., V, $\theta^*$). Missing information can be understood by a domain expert, but the paper should be self-contained as much as possible if not citing a prior work. Also, many statements are over-generalized.
>
> $\textbf{A3:}$ Thanks for the constructive suggestion. We have revised the draft and rebuilt the paper in this submission. We will try to list all the improvements and revisions in the final review abstract if it is available.
>
> ----
>
> $\textbf{Q4: [How the gradient updating work]}$ If the trigger does not appear in the training data, the embedding still wouldn't be updated as Q is computed and applied per sentence according to Eq 4. So, it is unclear how this update is significantly different from just updating the token with its own gradient only. Comparing the gradient and Q can be interesting. It almost looks like the effect is using a larger learning rate, but there was not sufficient analysis on this.
>
> $\textbf{A4:}$ Yes, this problem is related to the minor issue in Eq.(4). In the corrected formulation (of the re-submission version), we compute Q for each token in the vocab (including common tokens and rare tokens). Even if the rare tokens never appear in the input batch, the rare tokens' embeddings will also be updated with the pulling force Q. In Section 5.2 (of the re-submission version), we provide a more detailed analysis between Q and the standard gradient update and testify its effectiveness in erasing the triggers.
>
> ----

---

> > ### Author Response · Authors · 2021-12-02
> > **Follow Up**
> >
> > Dear reviewer,
> >
> > Do you still have any concerns about our manuscript? We are sincerely looking forward to your further feedback!

---

### Decision · Program_Chairs · 2022-01-20

**Decision:**

Reject

**Comment:**

This paper introduces a defense method (gradient broadcast adaptation) against backdoor attacks on pretrained language models. It proposes to utilize prompt tuning to guide the perturbed weights back to a normal state and thus helps avoid the degradation of model's generalization ability.

Strengths:
- Experiments are conducted across multiple datasets with different types of backdoor attacks, demonstrating the effectiveness of the proposed approach
- The proposed idea is well motivated and intuitive

Weakness:
- Improvement on experiment results seems marginal
- Some technical details of the attack setup are unclear
- Writing of the paper needs improvement